# Any-Start-Time Planning for SIPP

**Devin Wild Thomas,**[1] **Solomon Eyal Shimony,**[2] **Wheeler Ruml,**[1] **Erez Karpas,**[3]
**Shahaf S. Shperberg,**[2] **Andrew Coles**[4]

[1]University of New Hampshire, USA, [2]Ben-Gurion University of the Negev, Israel
[3]Technion, Israel, [4]King's College London, UK
dwt@cs.unh.edu, shimony@cs.bgu.ac.il, ruml@cs.unh.edu, karpase@technion.ac.il,
shperbsh@post.bgu.ac.il, andrew.coles@kcl.ac.uk

## Abstract

The problem of navigation among moving obstacles, some-times referred to as SIPP, is a problem in which the applica-bility of an action, such as moving to a particular neighboring location, can change with the passage of time. This means that the optimal plan, and its duration, can change depend-ing on when execution begins. In practice, the execution start time is often unknown until planning completes or another agent gives the go-ahead. However, most prior work on SIPP assumes a known start time and thus that the found plans are always feasible. In this paper, we relax this assumption and directly address the setting of any-start-time planning. We present a general-purpose data structure that compactly en-codes the optimal plan as a function of start time, as well as planning algorithms that utilize it. In experiments on SIPP, we find that this data structure tells us how long our paths re-main valid, while its overhead is minimal. We also find that, in any-start-time SIPP, replanning is insufficient for difficult problems. In contrast, our any-start-time algorithms can be quickly queried for the optimal plan once the start time is known, albeit at the cost of substantial precomputation time.

## Introduction

Many applications of planning naturally involve planning on a time-dependent graph. Often this time dependence can be represented as safe intervals within which a state or action is applicable or safe for the agent. This paper explores how a simple change to this problem, making the start time un-known, fundamentally changes how we address the prob-lem. To illustrate, consider a food delivery robot planning its delivery route. The robot must wait for a human chef to finish preparing a meal before it can be delivered. Assume we have accurate (enough) knowledge of the travel times on the two relevant local roads, such as when trains block crossings. If we know ahead of time when the food will be ready, i.e., the execution start time, we can find the optimal plan assuming departure at that time. This can be done by searching our road network for the earliest possible arrival time at each location, until we reach the goal. This problem representation, and a popular algorithm to solve it, are called Safe Interval Path Planning (SIPP) (Phillips and Likhachev 2011).

In more detail, Figure 1 shows the arrival time plotted as a function of departure time for the delivery robot. The robot

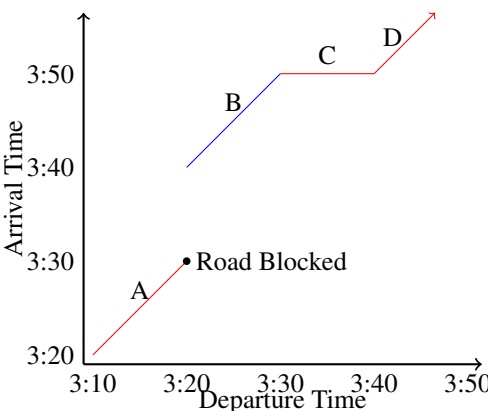

Figure 1: Delivery robot arrival time function, red segments follow the faster road, blue the slower road.

has two roads it can choose between, taking 10 and 20 min-utes, respectively, to reach the delivery target. However, the faster road is blocked from 3:20 PM - 3:40 PM by a freight train. We can see that there is a family of trajectories, la-beled as segment A in Figure 1, which represents the same journey along the faster road, just with different departure times. However, the train blocking the faster road causes a breakpoint in the arrival time function, where the segment labeled B corresponds to the delivery robot taking the slower road. At 3:30 there is another breakpoint in the arrival time function, where segment C has a constant arrival time, be-cause it is faster for the delivery robot to wait for the train to pass and then take the faster road. At 3:40 the train cross-ing opens and the arrival time function of segment D returns to increase proportionally to departure time. This structure arises due to the binary nature of safety in our problem.

### Relation to Previous Problem Settings

Prior work on time-dependent shortest paths (TDSP) (Drey-fus 1969; Orda and Rom 1990; Foschini, Hershberger, and Suri 2014) has shown that the arrival times of the optimal paths for the robot form a piecewise linear function. Each segment of the function represents a family of related paths going through the same locations but shifted in time. Using the arrival time functions (ATFs) for planning we will gen-

| start time | time passes during planning | |
| --- | --- | --- |
| | no | yes |
| known | temporal planning | contract search |
| unknown | **any-start-time planning** | situated planning |

Table 1: Relationships between problem settings.

eralize SIPP to Any-start-Time SIPP (@SIPP).

Any-start-time planning fills a gap between existing problem settings (Table 1). Returning to our delivery robot example, if we knew in advance when the chef will finish cooking and planning was fast enough to treat the problem as offline, we could model the situation as a conventional temporal planning problem with timed initial literals (Cresswell and Coddington 2003; Edelkamp and Hoffmann 2004; Fox and Long 2003). If planning is slow enough to require taking its duration into account, we have the contract search setting, in which the start time functions as a deadline by which planning must complete (Dionne, Thayer, and Ruml 2011). However, in many cases we are uncertain of exactly when we will be able to start executing any delivery plan that we make, so finding a single plan in advance will not work. Another approach would be to begin planning as soon as the food is ready. If planning is slow relative to the changing travel times and changing applicable actions, one could use a situated temporal planner (Shperberg et al. 2021; Cashmore et al. 2018) that takes the passage of time during planning into account during the planning itself in order to ensure that the final plan is feasible. The remaining combination is planning in the offline setting, but where we do not know the start time. A simple approach would be to assume that planning is so fast, and the environment changing so slowly, that we can just do temporal planning starting the moment we learn the start time. This however wastes time, and may result in an invalid plan if the planner is too slow.

Note that our example provides time before the food is ready during which the delivery agent can plan. The first contribution of this paper is a data structure from which the optimal plan can be quickly extracted once the start time is known. That is, once the food is ready, delivery can commence immediately without requiring any further search.

Related representations have been used to address TDSP problems in settings such as network routing (Orda and Rom 1990) and transit networks (Sancho 1992). Any-start-time planning is an example of the time-dependent shortest path problem with restricted slopes, which is noted by Foschini, Hershberger, and Suri (2014) due to the theoretical linear complexity of the ATF. We use the term any-start-time planning for brevity and to be clearer to a planning-focused audience. To our knowledge, this is the first work to instantiate a search algorithm that takes advantage of these theoretical complexity results. In this paper we 1) describe any-start-time planning, 2) specify the @SIPP problem, its relation to the TDSP problem and present PEAT, an algorithm to solve @SIPP, and 3) present the augmented SIPP algorithm, that captures useful extra information with minimal overhead. In the following sections, we show how the ATFs that were used by Foschini, Hershberger, and Suri (2014) to prove the

complexity of the TDSP naturally arise in the SIPP problem setting and how they can be adapted to form the states of a search algorithm to solve @SIPP.

## Background

A SIPP problem is a tuple $\langle S, E, \delta, x_o, x_g \rangle$. For every state $s = \langle x, i \rangle \in S$, the component $x = \langle x_0, x_1... \rangle$ denotes the configuration the robot is in during that state. In our delivery robot example this could be the location of the robot. The component $i = \langle t_s, t_e \rangle$ defines a safe interval, a continuous set of times from $t_s$ to $t_e$ where the corresponding configuration is safe. Thus, each state represents an interval of time that a configuration is safe. Many states may be associated with the same configuration, but with different intervals. We require that states have maximal intervals, that is that the configuration is unsafe immediately prior to the start of the interval, and immediately after the end of the interval.

An action $e = \langle u, v, i \rangle \in E$ represents an interval in time that the transition from state $u$ to state $v$ is safe for the agent[1]. We denote the end of the interval at state $u$ as $t_e^u$, while $t_s^e$ is the start of the interval of edge $e$. The cost of an action is its non-negative duration $\delta(u, v)$: an agent departing $u$ at time $t$ will arrive at $v$ at $t + \delta(u, v)$. Wait actions $\langle u, u, i \rangle$ may have any non-negative duration, subject to the constraints of the safe interval of $u$. All other actions have a time-independent constant duration. As with states, the safe intervals of edges are required to be maximal. The existence of an action implies that there exists a time $t$ such that the source is safe, the action is safe, and the destination is safe at $t + \delta(u, v)$. Our objective is to find a shortest path to the goal configuration $x_g$, from starting in the initial configuration $x_0$ at time 0.

The SIPP state space models an omniscient agent with complete knowledge of its environment, including moving obstacles now and in the future. These moving obstacles are represented in the state space as safe intervals at configurations and for actions moving between them. The fundamental observation for solving SIPP is the temporal dominance relation: it is always at least as good to be in a state earlier. This holds because any later time in an interval is reachable by waiting. The SIPP algorithm exploits this by tracking the earliest arrival time at each state as $g$ in the nodes of an A* search (Phillips and Likhachev 2011). This efficiently restricts the search to the compact SIPP state space rather than the infinite $\langle x, t \rangle$ state space inherent to these problems.

## Any-start-time SIPP

We define any-start-time SIPP (@SIPP) similarly to SIPP, with the sole difference that the initial start time $t_0$ is unknown. With this modification, we must now find a shortest path for *any* start time. In the food delivery example, the aim is to find delivery plans for all possible start times such that a plan can be immediately executed as soon as the food is ready. A side effect of allowing any start time is that, while

---

[1]We explicitly define actions as having safe intervals. Prior literature on SIPP has action intervals implicitly defined by collision checking. For example: if we have an agent on a 1-D grid, with an obstacle approaching from the right, the safe interval if the agent is moving to the right is different than if it is stationary.

the SIPP dominance relation still holds, it does not suffice to track only earliest arrival time for @SIPP. For example, imagine our delivery robot with two potential roads. The first is shorter, but crosses a set of train tracks and is only safe if the agent departs in the next 5 seconds. The second route is slightly longer, but without any anticipated interruptions. Searching only with regard to earliest arrival time would prune all information about the second route, and produce a plan that is only safe for the next 5 seconds. We need to track an ATF that encodes the earliest arrival time for all potential departure times. In the following section we describe how to represent these ATFs and how to use them when searching in the @SIPP state space.

## Representing Any-Start-Time Plans

Any safe path to the goal has up to two distinct sections to its ATF, for example sections C and D in Figure 1. The first is a constant section, where the path requires the agent to wait for some duration, and so departing later simply results in less waiting. Following this, there is then a breakpoint where the slope of the piecewise linear function changes. The path no longer requires waiting and any delay in departure would result in a proportionally later arrival, which introduces an increasing section where the ATF increases directly proportional to the departure time. For example, when our delivery robot is forced to wait at the train crossing, the departure time does not change the arrival time until the crossing becomes safe. As Figure 1 illustrates, an ATF representing multiple paths can have many segments.

Our key observation is that our search can use these ATFs analogously to $g$ values in SIPP, finding a path that represents a family of related valid traversals of the graph instead of the single traversal that would be found by conventional known-start-time SIPP algorithms. To do this, each search node keeps an ATF representing a set of partial paths the agent might take. Like $g$, we update this ATF when generating successor states. Each additional action can potentially constrain the entire path, requiring the agent to wait to depart for longer, until the action becomes safe, or setting an earlier deadline on the agents departure so that it takes the action before it becomes unsafe. We will first construct a finite directed graph representation of our problem, adapting the structure of TDSP problems. Second, with this representation, we will define the ATF of an edge on the graph, $A[e](t)$. Third, we then show how to construct the ATF of a path, $A[p](t)$ from its constituent edge ATFs. Fourth and finally, we explain how to represent a set of path ATFs and search using ATFs.

## Earliest ATFs for Edges

We will now address how to manipulate our @SIPP state space into a finite directed graph analogous to those used in the TDSP literature. We adapt the notation for ATFs from Foschini, Hershberger, and Suri (2014). To generate this graph, we observe that it is possible to compile the state intervals into the respective action intervals. Precisely, we construct a finite directed graph, $G = (S, E, A[e])$ from an @SIPP problem $\langle S, E, \delta, x_0, x_g \rangle$. The vertices of the graph are the states of the @SIPP problem, likewise the edges are

the actions. The change we make is to compile the safe intervals of the source state, destination state, and the action into one ATF, $A[e]$. As we saw in the delivery example, $A[e]$ encodes the inherent minimum duration to travel between the two configurations, while also accounting for any waiting required by the intervals of the source, destination and action.

To compile an action $e = \langle u, v, i \rangle$, we parameterize $A[e]$ as three time points and a duration, these parameters are illustrated in Figure 2a. "wait", denoted $\zeta$, is $t_s^u$, the earliest departure time that the agent can begin waiting at the source state $u$. The second time is "go", denoted $\alpha$, the earliest departure time to traverse $e$. $\alpha$ is a breakpoint we described earlier, for example the breakpoint between C and D in Figure 1. $\alpha$ is constrained by all three intervals of $e$, $u$ and $v$ because the agent must be safe in $u$ and safe to traverse $e$ at the moment of departure, and also be safe upon arrival in $v$ at $t_{depart} + \delta(u, v)$. The precise calculation of $\alpha$ is shown in Eq. 2. The final time, "end" denoted $\beta$ is the latest departure time that will allow a safe traversal of $e$. Like $\alpha$, $\beta$ is constrained by all three intervals; the calculation is shown in Eq. 3. The duration $\Delta$ is $\delta(u, v)$.

$$\zeta = t_s^u \tag{1}$$
$$\alpha = max(t_s^e, t_s^u, t_s^v - \delta(u, v)) \tag{2}$$
$$\beta = min(t_e^e, t_e^u, t_e^v - \delta(u, v)) \tag{3}$$
$$\Delta = \delta(u, v) \tag{4}$$

These parameters define the piecewise linear edge ATF:

$$f[\zeta_e, \alpha_e, \beta_e, \Delta_e](t) = \begin{cases} \infty & t < \zeta_e \\ \alpha_e + \Delta_e & \zeta_e \le t \le min(\alpha_e, \beta_e) \\ t + \Delta_e & \alpha_e \le t \le \beta_e \\ \infty & \beta_e < t \end{cases} \tag{5}$$

If the agent 'departs' $u$ between $\zeta$ and $\alpha$, the soonest it can arrive at $v$ is by waiting at $u$ until $\alpha$ and then traversing the edge, arriving at time $\alpha + \Delta$, where $\Delta$ is the traversal duration of edge $e$, regardless of when in the $\zeta - \alpha$ window the agent departs $u$. After $\beta$, either the edge or $v$ becomes inaccessible and no traversal is possible, so $A[e] = \infty$ again. If the agent arrives at $u$ at some time $t$ between $\alpha$ and $\beta$, it can transit immediately to $v$ without waiting, arriving there at $t + \Delta$. $A[e]$ captures all the information needed from the original intervals on $u$, $v$ and $e$. We note that conditions of the second and third terms of Eq. 5 may appear odd, this is because we will later reuse the same piecewise linear parametrization for the ATF of a path. We need $min(\alpha, \beta)$ because the ATF of a path may have $\beta < \alpha$ when traversing the path requires more waiting ($\alpha$) than can be done in the initial state ($\beta$). In such a case $\beta < \alpha$ means the third case, $\alpha \le t \le \beta$ never applies. We will explain this further after we describe path ATFs in the following subsection.

## Earliest ATFs for Paths

We need two pieces of information to generate a successor, $A[e]$ is the first, filling the role that action duration would normally play in SIPP. Second is the path ATF $A[p]$, that acts as $g$ in our search. As we have replaced $g$ and duration

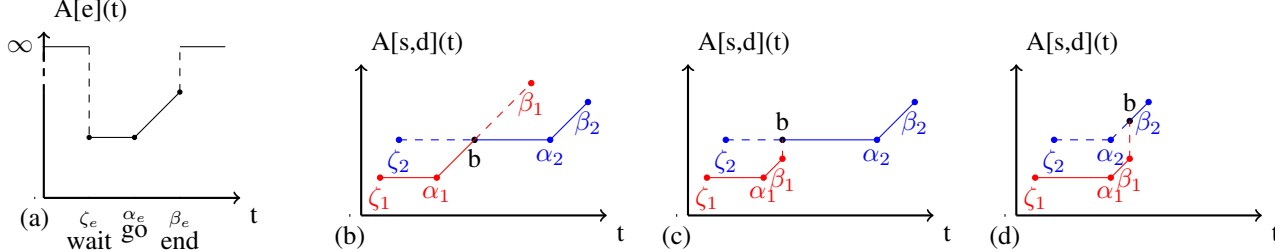

Figure 2: (a) A typical A[e] (b) – (d) Examples of the interaction of two path ATFs $A[p_1]$ and $A[p_2]$ in forming $A[s,d]$

with ATFs, we can no longer simply add the incremental duration to update $g$ for a newly generated successor. It may be helpful to view the search as building out paths, rather than searching through states. Using the edge earliest ATFs, we now define ATFs for paths: $A[p]$. A path represents an ordered subset of edges where it is safe to traverse each edge in order, possibly with some waiting. So intuitively, the ATF of a path is the successive composition of the ATFs of the edges in the path. The departure time of each succeeding edge is the earliest arrival time of the preceding edge. What may be surprising is that the composition operation maintains the piecewise linear structure of the edge ATFs, meaning that we can represent a path ATF by its equivalent edge ATF. Next, we will see how this composition maintains the simple structure of the ATF.

The ATF for a feasible path $p$, denoted by $A[p](t)$ which we restate from Foschini, Hershberger, and Suri (2014) is:

$$A[p](t) = A[p_n] \circ A[p_{n-1}] \circ \ldots \circ A[p_1](t) \qquad (6)$$

with $\circ$ denoting function composition: $(f \circ g)(t) = f(g(t))$. $A[p](t)$ is the earliest arrival time at the end of $p$, following path $p$, starting at time $t$. The agent might be able to do additional waiting along the path without invalidating the path, although it would arrive later.

We observe that the path ATF of a path composed of a single edge, is exactly the ATF of that edge. This leads us to introduce the representation of $A[p]$ as its 'equivalent edge' with regards to the ATF. The role of an equivalent edge is to stand in for a path in computing the ATF, allowing us to use Eq. 5 to compute $A[p]$ as well as $A[e]$. That is, we want to find the parameters of an equivalent edge $e_{eq} = \langle \zeta, \alpha, \beta, \Delta \rangle$ such that the edge ATF is exactly the path's ATF.

Consider a path consisting of three edges, the first safe from $[0, 2]$, the second from $[0, 10]$ and the third from $[8, 10]$ each with unit movement duration. The agent can wait for a maximum of two time units before taking the first edge, so the $\beta$ of the equivalent edge is 2. However the agent is forced to wait for at least 6 time units for the final edge to become accessible, so $\alpha$ of the equivalent edge is 6. Thus our equivalent edge is parameterized by $\langle 6, 2, 3 \rangle$, giving us an earliest arrival time of $6 + 3 = 9$, which increases proportional to departure time for a latest arrival time of $6 + 2 + 3 = 11$.

Notice that this example shows a case where $\alpha > \beta$, which requires $min(\alpha, \beta)$ in Eq. 5. For a single edge, all waiting must be done prior to taking the action to traverse the edge. However, for a multi-step path, the agent can

sometimes wait at intermediate points. While we can not be forced to wait for longer than the edge is safe, it can happen that we are forced to wait along an entire path for longer than we can wait prior to traversing the first edge! Precisely, while $\alpha \leq \beta$ is true for all edges, it need not be true for a path. This arises when the agent is forced to wait along the path but cannot do all the waiting at the beginning. Instead, the agent must wait at intermediate states along the path.

Formally, to generate the successor $p'$ of path $p$ using action $e$, the ATF of $p'$ is:

$$A[p'](t) = A[e] \circ A[p](t) \qquad (7)$$

As the base case, suppose we have a path composed of a single edge: the ATF of the path will be exactly the ATF of its single edge. Now let us examine the ATF of a path with two edges: $p$, the equivalent edge of the existent path; and $e$, the edge being composed onto the path. $\zeta_{p'}$ is exactly $\zeta_p$, as extending the path does not change the safety of its start. Each edge along the path can either force the agent to wait for it to open, or not place an additional requirement on the minimum waiting that an agent must do along the path. Similarly, the expiration of each edge provides a deadline on the most additional waiting that can be done at each step along the path. The intrinsic duration of movement in the path, $\Delta$, is the sum of the intrinsic durations of the edges. We formalize these intuitions into a recursive definition of our equivalent edge:

$$\zeta_{p'} = \zeta_p \qquad (8)$$
$$\alpha_{p'} = max(\alpha_p, \alpha_e - \Delta_p) \qquad (9)$$
$$\beta_{p'} = min(\beta_p, \beta_e - \Delta_p) \qquad (10)$$
$$\Delta_{p'} = \Delta_e + \Delta_p \qquad (11)$$

With this we now have all we need to generate successors using $A[p]$ in place of $g$ and $A[e]$ in place of duration.

## Multi-path Earliest ATFs

The last requirement for search is an ATF for a set of paths, all with the same source state $s$ and destination $d$. We denote the ATF for reaching $d$ from $s$ as $A[s, d]$. When planning, we will be considering multiple possible paths. If $\mathcal{P}_{s,d}$ represents the set of all feasible paths from $s$ to $d$, then $A[s, d]$ minimizes over these (restated from Foschini, Hershberger, and Suri (2014)):

$$A[s, d](t) = \min_{p \in \mathcal{P}_{s,d}} A[p](t) \qquad (12)$$

As $A[s,d](t)$ is a minimization over piecewise linear $A[p](t)$ path arrival functions, $A[s,d](t)$ is also piecewise linear. Figure 2b, c, d shows different combinations of breakpoints that can arise in $A[s,d]$. The breakpoints of each path ATF $A[p]$ are labeled $\zeta_i$, $\alpha_i$ or $\beta_i$. Regions minimizing $A[s,d]$ are shown as a solid line while the remainder of the path ATF is dashed. Minimization breakpoints are black dots labeled $b$. Discontinuities in $A[s,d]$ are highlighted with a dashed line. Figure 2b shows the unique configuration leading to a distinct breakpoint $b$ that was not originally a breakpoint of either constituent path. Figure 2c,d show breakpoints in $A[s,d]$ that are not distinct from the path breakpoints that cause them.

## Searching with ATFs

We now have what we need to search with ATFs. We illustrate their use via the operation of A* on the delivery robot scenario. We represent the @SIPP problem using three states: the restaurant $\mathcal{R}$, which has a safe interval during dinner service of $\langle 2{:}00, 8{:}00 \rangle$, the railroad crossing $\mathcal{C}$, and the apartment $\mathcal{A}$ to deliver the food to. The start state is $\mathcal{R}$, the goal is $\mathcal{A}$, our search must compute $A[\mathcal{R}, \mathcal{A}]$, which is shown in Figure 1. The crossing and apartment are always safe. The action moving from $\mathcal{C}$ to $\mathcal{A}$ has a safe interval representing the train. Formally, the actions are:

$$e_1 = \langle \mathcal{R}, \mathcal{C}, \langle -\infty, \infty \rangle \rangle$$
$$e_2 = \langle \mathcal{C}, \mathcal{A}, \langle -\infty, 3{:}20 \rangle \rangle$$
$$e_3 = \langle \mathcal{C}, \mathcal{A}, \langle 3{:}40, \infty \rangle \rangle$$
$$e_4 = \langle \mathcal{R}, \mathcal{A}, \langle -\infty, \infty \rangle \rangle$$

The inherent travel times are:

$$\delta(\mathcal{R}, \mathcal{C}) = 1 \text{ min}$$
$$\delta(\mathcal{C}, \mathcal{A}) = 9 \text{ min}$$
$$\delta(\mathcal{R}, \mathcal{A}) = 20 \text{ min}$$

We now calculate the parameters $\langle \zeta, \alpha, \beta, \Delta \rangle$ of our edge ATFs, using Equations 1-4:

$$A[e_1] = \langle 2{:}00, 2{:}00, 8{:}00, 1 \text{ min} \rangle$$
$$A[e_2] = \langle -\infty, -\infty, 3{:}20, 9 \text{ min} \rangle$$
$$A[e_3] = \langle -\infty, 3{:}40, \infty, 9 \text{ min} \rangle$$
$$A[e_4] = \langle 2{:}00, 2{:}00, 8{:}00, 20 \text{ min} \rangle$$

Search nodes consist of a state (for clarity, we list the full path) and its path ATF (taking the place of the traditional $g$ value). For this example, we will use $h(n) =$ the cost to go assuming all states and actions are always safe. A*'s open list will be sorted by the earliest expected arrival time ($f(n) = \alpha(n) + \Delta(n) + h$).

For our starting node, the path is simply $\mathcal{R}$, our ATF is uninformed so far, and $f$ is $-\infty + 0 + 10$:

$$\text{open}_0 = \{\langle \mathcal{R} : \langle -\infty, -\infty, \infty, 0 \text{ min} \rangle, f = -\infty \rangle\}$$

We pop the front of open, and generate the successors of $\mathcal{R}$. To do so, we look at the actions, $e_1$ and $e_2$ originating in

$\mathcal{R}$. Each successor is generated by applying Equations 1-4 yielding two successors:

$$\text{open}_1 = \{\langle \mathcal{RC} : \langle 2{:}00, 2{:}00, 8{:}00, 1 \text{ min} \rangle, f = 2{:}10 \rangle,$$
$$\langle \mathcal{RA} : \langle 2{:}00, 2{:}00, 8{:}00, 20 \text{ min} \rangle, f = 2{:}20 \rangle\}$$

Popping the front of open, we expand $\mathcal{RC}$, yielding two more successors:

$$\text{open}_2 = \{\langle \mathcal{RCA}_1 : \langle 2{:}00, 2{:}00, 3{:}19, 10 \text{ min} \rangle, f = 2{:}10 \rangle,$$
$$\langle \mathcal{RA} : \langle 2{:}00, 2{:}00, 8{:}00, 20 \text{ min} \rangle, f = 2{:}20 \rangle,$$
$$\langle \mathcal{RCA}_2 : \langle 2{:}00, 3{:}39, 8{:}00, 10 \text{ min} \rangle, f = 3{:}50 \rangle\}$$

Open is now $A[\mathcal{R}, \mathcal{A}]$, very similar to what is shown in Figure 1. The slight difference is due to the original example used by Figure 1 not having the state $\mathcal{C}$, which we added to make this worked example more illuminating. Segment A is $\mathcal{RCA}_1$, segment B is $\mathcal{RA}$ and C and D are $\mathcal{RCA}_2$. Our next steps would be to pop the three nodes off open, observe that they have reached the goal, and add them to our $A[s,d]$ container. We can then query $A[s,d]$ for a specific starting time, for instance if the robot departs at 3:18 the minimizer of Equation 12 is $\mathcal{RCA}_1$, and we have up to 1 minute to safely depart because $\beta = 3{:}19$.

## Computing $A[s,d]$ Efficiently: Implementation

In order for $A[s,d]$ to be useful, we would like to have a representation that can be efficiently queried to return the optimal path for a given time. It should also be efficient to incrementally construct during a search. We represent $A[s,d]$ as a set of line segments, with each constituent $A[p]$ forming a pair of segments inserted into $A[s,d]$. Each segment is parameterized by $\langle begin, end, slope, a_b \rangle$ where $begin$ and $end$ delineate the extent of the segment in departure time, $slope$ is the slope of the segment which is either zero or one and $a_b$ is the arrival time of the segment at $begin$. We can represent $A[p] = \langle \zeta, \alpha, \beta, \Delta \rangle$ as segments $\langle \zeta, \alpha, 0, \Delta \rangle$ and $\langle \alpha, \beta, 1, \Delta \rangle$. These segments can be stored in a balanced tree sorted on $\beta$, with a pointer to $A[p]$. This structure gives $O(log(n))$ query and add. We note that if needed it may be possible to effectively get constant time query for the current time, by maintaining a pointer to the $A[p]$ corresponding to the current time.

We add a prospective $A[p]$ to $A[s,d]$ when $A[p]$ is not dominated by $A[s,d]$, which is when there exists some time such that $A[p](t) < A[s,d](t)$. Because $A[s,d]$ is monotonic non-decreasing, it is sufficient to check for dominance at the breakpoints of $A[p]$ and $A[s,d]$.

## Planning Algorithms

We present three algorithms that we will use to study search with ATFs: one to compare against SIPP at solving SIPP problems and two for solving @SIPP problems:

**ASIPP** Augmented SIPP: performs an identical search as SIPP but with $A[p]$ rather than scalar $g$.

**RSIPP** Replanning SIPP: finds an approximate solution to @SIPP by precomputing an initial SIPP plan for $t = 0$, then replanning while invalid.

**Algorithm 1: SIPP**

```
 1: function SIPP(startState, goalConfig)
 2:     open ← {}, closed ← {}
 3:     place startState on open and closed
 4:     while open not empty do
 5:         cur ← open.pop()                    ▷ min f
 6:         if cur at goalConfig then
 7:             return path of cur
 8:         for e ∈ successors(cur) do
 9:             dt ← δ(cur.state, e.v)
10:             g ← max(cur.g + dt, t_s^e)
11:             n ← ⟨e.v, g⟩
12:             if n ∉ closed or n better then
13:                 put n in open and closed
14:     return Failure
```

**Algorithm 2: RSIPP**

```
15: function RSIPP(startConfig, goalConfig)
16:     validPlan ← False
17:     Wait until t_start
18:     while true do
19:         p ← SIPP(⟨startConfig, t_now⟩, goalConfig)
20:         if p still valid then
21:             return p
```

**PEAT** Partial Expansion A* for @SIPP: solves @SIPP by continuing a partial expansion ASIPP and maintaining an arrival time function.

These three algorithms and SIPP all operate on a search graph $G = (S, E, A[e])$.

Algorithm 1 formally shows how SIPP performs an A* search. We perform the search, popping the minimum $f$ node off open. If the node is a goal, we follow the parent pointers of the node to recreate the path. Otherwise, we generate the successors, which are the edges in $G$ that start in $cur.state$ (and are safe at $cur.g$). In line 9 we calculate $dt$, the inherent duration of action $e$, and in line 10 we calculate the (potential) earliest arrival time as the maximum of $cur.g + dt$ which is the arrival time if we are able to depart at the earliest arrival time at $cur$, otherwise we must wait for the edge to become safe at time $t_s^c$. We store these values as the node $n$, then in line 12 if $n.state$ has not been generated before or if we have found an earlier arrival time at $n.state$, we add $n$ to open and closed.

Augmented SIPP (Algorithm 3) modifies SIPP to perform an identical search, solving a SIPP problem but returning $A[p]$ rather than a single path. This is done by lines 31-33 which update $A[p]$. Because ASIPP returns $A[p]$, we will know not only the earliest arrival time at our destination, but $\alpha$ tells us how much time we will be forced to wait along the way and $\beta$ tells us when the path we have planned becomes unsafe. These may be valuable in certain applications. For example, imagine our delivery robot knows it will be forced to wait near its destination. Earlier along the path, it now has the freedom to not arrive at the earliest arrival time at each intermediate state. This allows it to know that it has time to

**Algorithm 3: Augmented SIPP**

```
22: function ASIPP(startState, goalConfig)
23:     open ← {}, closed ← {}
24:     place startState on open and closed
25:     while open not empty do
26:         cur ← open.pop()                    ▷ min f
27:         if cur at goalConfig then
28:             return path of cur
29:         for e ∈ successors(cur) do
30:             dt ← δ(cur.state, e.v)
31:             α ← max(e.α - cur.Δ, cur.α)
32:             β ← min(e.β - cur.Δ, cur.β)
33:             Δ ← cur.Δ + dt
34:             n ← ⟨ nC, dest, α, β, Δ ⟩
35:             if n ∉ closed or n better then
36:                 put n in open and closed
37:     return Failure
```

drive in a more conservative manner, potentially making the plan more robust to unforeseen events.

Our second algorithm, RSIPP, approximates a solution to @SIPP, where the agent is planning for an unknown start time. RSIPP waits until the start time, then repeatedly runs SIPP starting from the current time $t_{now}$. If the found plan is still valid after planning completes, we return it, otherwise loop. If the problem is simple enough and the planner fast enough, this should return a valid path in time to be used. It however does not truly solve the @SIPP problem, as we are only returning a single path.

In contrast, PEAT (Algorithm 4) maintains a collection of paths to the goal that it progressively adds to, increasing the departure time it has plans for. PEAT starts in line 40 by initializing open with the starting configuration at time 0, which is a stand-in for the earliest departure time we might query. @SIPP is potentially an infinite sized problem as we are concerned with finding the paths at all times including infinitely far in the future. This creates an issue where a node may (in theory) have an infinite number of successors. While we want to have a solution for any start time, time is monotonically increasing and we generally want solutions for departure times in the near future before solutions for the far future. To address this, PEAT performs a partial expansion A* (Yoshizumi, Miura, and Ishida 2000), where we generate only a subset of successors each time a node is expanded, corresponding to the earliest upcoming safe intervals at neighboring configurations which have not already been generated. We pop the minimum $f$ node from open in line 42, then in line 45 we generate the next layer of successors of $cur$, and finally in line 55 we return $cur$ in open, with its expansions count incremented. Concretely, $cur$ tracks the number of times it has been expanded, so on the first time it is expanded, it generates successors for the chronologically first edge to each successor configuration, on the $n^{th}$ time it is expanded it generates the successors corresponding to the $n^{th}$ edge to each successor configuration. In our delivery robot example, the first time $\mathcal{RC}$ was expanded it would generate the successor from following edge $A[e_2]$, then be

| Algorithm 4: PEAT |
| --- |

```
38: function PEAT(startConfig, goalConfig)
39:     open ← {}, closed ← {}, A[s, d] ← {}
40:     place ⟨startConfig, 0⟩ on open and closed
41:     while open not empty do
42:         cur ← open.pop()                    ▷ min f
43:         if cur at goalConfig then
44:             add path of cur to A[s, d]
45:         for e ∈ nextSuccessors(cur) do
46:             dt ← δ(cur.state, e.v)
47:             α ← max(e.α - cur.Δ, cur.α)
48:             β ← min(e.β - cur.Δ, cur.β)
49:             Δ ← cur.Δ + dt
50:             n ← ⟨ nC, dest, α, β, Δ ⟩
51:             if n ∉ closed or n better then
52:                 put n in open and closed
53:         cur.expansions += 1
54:         cur.f = min f of remaining children
55:         place cur in open
56:     return A[s, d]
```

placed back onto open. The second time $\mathcal{RC}$ is expanded, it would generate the successor from following edge $A[e_3]$ and, because it has no more successors, $\mathcal{RC}$ would not be placed back on open.

The second key element of PEAT is that it maintains $A[s, d]$, the record of found paths to the goal. $A[s, d]$ is initialized empty, then rather than returning when a path to the goal is found, we add it to $A[s, d]$ in line 44. The pruning strategy for PEAT remains an area of future work. An improved version of PEAT would prune nodes that represent plans that are *worse* than plans already in $A[s, d]$, but should still allow *short duration* plans that depart *later*.

## Empirical Evaluation

To evaluate the performance of these techniques, we ran SIPP experiments with 8-way motion by both agent and obstacles. We address three questions: 1) what is the overhead of ASIPP in comparison with SIPP? This tells us the price we would be paying to know when our plan will become invalid. 2) How long do SIPP plans remain valid? This tells us when it would be worth using PEAT, in comparison to a simple replanning scheme like RSIPP. 3) What are the relative runtimes of precomputation and querying for RSIPP and PEAT? This tells us when an@SIPP vs replanning approach might apply. Our C++ code is available on GitHub[2].

### Experimental Set-Up

Our test domains are adapted from maps and scenarios from the Moving AI Lab 2D Pathfinding Benchmarks (Sturtevant 2012). We selected three maps, 32room_004, random512-20-1 and den520d (which we will refer to as rooms, random and den520d respectively) to provide three contrasting static environments. Rooms and den520d were selected, in part, to

[2]https://github.com/dwthomas/any-start-time-sipp

correspond with prior SIPP papers, which have used these or similar maps for their experiments. Rooms provides a static environment with regularly shaped and spaced local minima to increase the challenge of the search, while den520d consists of larger irregular rooms connected by long hallways. The random map contrasts with the other two, providing an intermediate difficulty between the easier den520d and the more challenging rooms. The den520d map is 256x257, while the other two maps are 512x512. We selected the 16 start-goal scenarios with the longest optimal paths in the static benchmark scenarios provided with each map.

For each map, we also generate 16 sets of 16,384 moving obstacles. Experiments are run on subsets consisting of the first $n$ of these obstacles, so experiments with more obstacles include those from the smaller experiments. Each obstacle moves from a chosen starting location by picking a direction and distance to move (or waiting in place). It moves in that direction until it hits a static obstacle, or travels the intended distance. This process is repeated to generate potentially infinite paths for the moving obstacles. The random generator is repeatable across different computers and compiler versions. Moving obstacles are able to pass through other moving obstacles, but not through the static environment. The obstacles are not following long distance optimal paths; for example in the rooms map, this means the obstacles spread out into the rooms, instead of streaming on shortest paths between doorways. This creates a more uniform distribution of safe intervals, rather than having high traffic paths surrounded by low traffic areas.

In order to generate safe intervals when an obstacle pauses at some grid point, we generate an unsafe interval at that grid position for the duration of the wait. When an obstacle moves between two grid points, the source location is recorded unsafe for the first half of the movement, the destination is recorded unsafe for the second half of the movement and the edge connecting those two locations is recorded unsafe for the entirety of the movement. A diagonal movement also blocks the other intersecting diagonal edge. Neither obstacles nor the agent can 'cut corners' of static obstacles. To constrain the problem to a finite size, all locations become unsafe after 5,000 seconds. All algorithms use octile distance as their heuristic.

The experiments were performed on identical machines with i3-12100 CPUs and 64 GB RAM. The algorithms were implemented in C++, sharing code where possible. All implementations share common data structures, including the node ordering for the open list, which prefers low $f$, breaking ties for higher $g$. The code was compiled by GCC v11.3.0 using -std=c++20 and -O3. All timings measure only the search itself, excluding generation of safe intervals from moving obstacles and I/O, which are similar for all the tested algorithms.

### Overhead of Augmented SIPP

Our first experiment measures the overhead of ASIPP in comparison with standard SIPP. We calculate the overhead as the difference in runtimes, normalized by the SIPP runtime: % overhead $= 100 * \frac{r_{asipp} - r_{sipp}}{r_{sipp}}$. We ran SIPP and

|       | 0 | 4 | 16 | 64 | 256 | 1024 | 2048 |       | 4 | 16 | 64 | 256 | 1024 | 2048 |
|-------|---|---|----|----|-----|------|------|-------|---|----|----|-----|------|------|
| rooms | 0.5±6 | 0.5±6 | 1.5±6.0 | 1.0±5 | 0.5±5 | 0.5±5 | 0.6±8 | rooms | 100 | 100 | 100 | 99 | 15 | 0 |
| den520 | 11±44 | 3.8±26 | 1.2±11 | 1.2±7 | 0.9±6 | 0.1±3 | 0.3±2 | den520 | 100 | 100 | 100 | 100 | 1 | 2 |
| random | 0.6±6 | 0.8±6 | 0.7±5 | 0.8±5 | 0.9±4 | 0.6±4 | 2.0 ±10 | random | 100 | 100 | 100 | 100 | 44 | 22 |

(a) Percent Overhead of Augmented SIPP          (b) Percentage of plans valid longer than SIPP runtime

Table 2: SIPP experimental results.

ASIPP on the three maps, with 0, 4, 16, 64, 256, 1024 and 2048 obstacles from each obstacle sets. Table 2a shows the average and standard deviation in percent overhead for each map and number of obstacles. In den520d we see a significant decrease in overhead from the instances with very few obstacles (11%) to more obstacles (0.3%). Because this is the smallest and easiest map when there are few obstacles, its overhead of 11% with high variability can be considered indicative of the worst case. Overall, we conclude that there is minimal runtime overhead associated with using Augmented SIPP to track the duration of path validity.

### How long are SIPP plans valid?

The family of plans found by ASIPP are valid from $t = 0 \ldots \beta$, and the plan found by SIPP is the earliest traversal of the ASIPP plan, so the $\beta$ found by ASIPP is the duration that the SIPP plan is valid when shifted in time. Table 2b quantifies the proportion of plans found that are valid for longer than the runtime of SIPP (confidence intervals are all are $< 1\%$ and are omitted). Detailed results are shown in Figure 3, where each point represents a single run. As more obstacles are included, there is both a decrease in the duration a plan is valid and an increase in runtime. For all three maps there is a significant decrease from 256 obstacles, where almost all plans are valid for longer than the runtime, to 1024 obstacles, where the majority of plans are invalid when the planner finishes. As 1024 is around 1.6% of a 256x257 map, we conclude that it does not take a particularly dense obstacle set to make an @SIPP approach attractive.

### RSIPP and PEAT runtime comparison

In order to test the runtime of PEAT, we ran PEAT on the 1024 obstacle instances and cut off planning after PEAT found optimal plans for at least the first 4 seconds, or it exhausted the search space. Four seconds was chosen to be beyond the average time where SIPP plans become invalid in these instances. On average, PEAT took 96 seconds to complete the precomputation, which is 40x the average runtime of SIPP. An average of 11 path ATFs were preserved in the final function, with a standard deviation of 8 and a maximum of 35. The small number of path ATFs suggests that querying $A[s, d]$ would be very fast. However, the pre-processing time is very long compared to solving a single SIPP problem with ASIPP, suggesting that PEAT would benefit from a more intelligent pruning strategy. One area of improvement we see in PEAT is that we are always searching with respect to a reference time of 0 (see Alg. 4.40). We are working on how to modify PEAT to search while monotonically increasing this reference time, efficiently finding paths that increase the time for which PEAT knows an optimal path.

## Related Work

Our work can be seen as a specialization of TDSP problems or as a generalization of SIPP. To our knowledge, while problems similar to @SIPP have been suggested in remarks by Halpern (1977) and Foschini, Hershberger, and Suri (2014), it has not been explored in prior work. Early work on TDSP by Cooke and Halsey (1966) and Dreyfus (1969) provided a Dijkstra style algorithm for finding the quickest path between nodes on a graph with time-dependent edge delays. Halpern and Priess (1974) introduced TDSP with arc closures, effectively safe intervals on edges. Later, Halpern (1977) introduced TDSP with parking bans on nodes, and time-dependent edge costs. However, Halpern (1977) and other more recent work are focused on the case where the node time windows are 'parking bans': where a node can be temporarily closed for waiting, but may still be traversed. They provide a manipulation to their graph that describes how to implement safe intervals on a node by replacing the node with a pair of nodes connected by a single edge corresponding to the safe interval. We note that the combination of the 'Nodes with no-passing-through periods' i.e. safe intervals described in Halpern (1977) and the edge functions from Halpern and Priess (1974) that alternate between a constant duration and infinite duration directly corresponds to our problem space.

Sancho (1992) provides a dynamic programming algorithm for the TDSP with time constraints on movement and parking described by Halpern and Priess (1974). Orda and Rom (1990) formalize the problem of Halpern (1977) and provide algorithms for a variety of extension and modifications to the problem. El-Sherbeny et al. (2014) give an A* algorithm for TDSP with time windows. Foschini, Hershberger, and Suri (2014) study the complexity of the problems presented in Orda and Rom (1990); they prove that for a restricted family of edge slopes, which includes our case of $\{0, 1, \infty\}$ the complexity of the ATF, $A[s, d]$, is linear in the number of edges of the graph. Modern work in TDSP has focused on rapidly answering repeated queries on a graph often using contraction hierarchies (Batz et al. 2009), or shortcuts (Delling 2011) as well as in logistics domains, for instance trucking with driving bans (van der Tuin, de Weerdt, and Batz 2018).

In standard temporal planning, time is considered but there is no exogenous change. Situated temporal planning (Shperberg et al. 2021) is limited by its duplicate state detection, designed for the general case of task planning, where TILs may have arbitrary state effects, and degrades to keeping all paths to all states on the special case of SIPP, where TILs are used only to demarcate intervals.

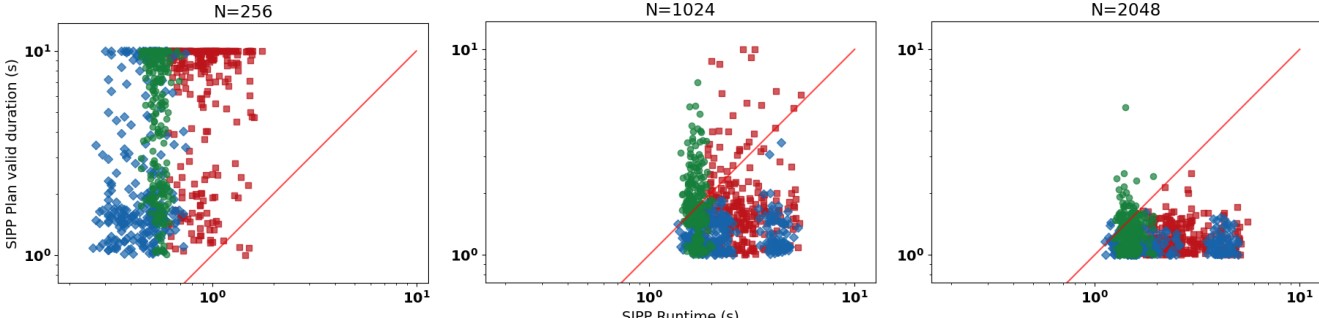

Figure 3: SIPP runtime vs. plan validity duration. Blue diamonds are experiments on the den520d map, green circles the random map and red squares the rooms map. The red line marks where the SIPP runtime is equal to the resulting plan's valid duration.

## Conclusion

Any-start-time SIPP complements the existing problem settings of temporal planning, situated planning, and contract search by formalizing a problem of offline planning for an unknown start time. This generalizes SIPP and is closely related to several variants of TDSP. The value of a state is no longer captured by a scalar but requires a function. Inspired by prior TDSP work, we showed that @SIPP can be solved using relatively simple and compact ATFs. We showed that the overhead of planning using ATFs is modest (at most 11% in our experiments). We also showed how they could be combined with partial expansion to yield an incremental any-start-time SIPP planner. This work strengthens the foundations of planning with time and exogenous change, two features that are important in many applications.

## Acknowledgments

This research was supported by United States-Israel Binational Science Foundation (BSF) grant 2019730, and United States National Science Foundation (NSF) grant 2008594.

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
