# OpenReview forum: "Any-Start-Time Planning for SIPP"
_icaps-conference.org/ICAPS/2023/Workshop/HSDIP — ICAPS HSDIP 2023_

### Official Review · Reviewer_VyKp · 2023-04-24
**Mostly easy to follow, but experimental evaluation could be more detailed**

**Rating:** 7
**Confidence:** 3

**Review:**

Paper summary: This paper generalizes Safe Interval Path Planning (SIPP) to any-start-time SIPP, enabling planning before a specific start time for the is known. To this end, it is demonstrated how the piecewise linear arrival time function (ATF) of a search path can be constructed inductively during search, and how the final ATF of the task can be constructed from the goal-reaching path ATFs using a data structure that supports efficient lookup of the corresponding path for any given input start time, once it is known.

I acknowledge the relevance of this paper for the workshop. Although this is definitely not my field of research, I would say that the paper makes a significant contribution. The central parts of the paper are fairly comprehensible, containing many examples and explanations that help following along with the developments made. The only exception for me was in Eq. 5, where I did not understand why the minimum of $\alpha_e$ and $\beta_e$ is used. For edge ATFs it seems redundant since here $\alpha_e < \beta_e$? One should mention this to avoid confusion. The construction of the earliest ATFs of edges, the inductive construction of the earliest ATFs for paths and the multi-path earliest ATFs are quite intuitive and correct to the best of my knowledge. Regarding the usage of ATFs during search, the provided example leaves little room for any misunderstandings, and up until this point, the paper is nicely written.

The search algorithms were a little less accessible to me. Namely, I am a bit unsure whether I fully understood how RSIPP operates in the any-start-time setting. In my understanding, RSIPP performs online planning by repeatedly computing a path from the current time point until the time spent while doing so did not already invalidate said path. However, that seems to be entirely different from the proposed any-start-time planning setting, but you also acknowledge that it does not "truly solve @SIPP", so perhaps I did understand it correctly.
ASIPP seems like a simple yet useful extension of the SIPP algorithm.

The experimental evaluation of the algorithms is okay. The ASIPP extension is shown to have only little overhead over SIPP, which is totally fine considering that ASIPP provides information that SIPP does not. I was a bit confused by the scatter plots in Figure 3, I do not really see a reason as to why the axes should be different. Also, what parameter is even changed between these three plots? If I had to guess, the plots show N=256, 1024, 2048 from left to right. The presentation should definitely be improved here. Nevertheless, what I get from this is that as the number of obstacles increases, the @SIPP becomes more suitable compared to replanning, which would take very long to find a path, or not find a path at all. Regarding PEAT vs RSIPP, 40x the runtime of SIP seems to be a lot considering that the algorithms are very similar. It is unfortunate that the evaluation is not more detailed at this point. It is suggested that the time loss does not come from the ATF data structure, which only leaves the larger search space of @SIPP as the reason for this big increase. It would be interesting to see how the number of expanded configurations and the overall pre-computation time for @SIPP scales with the number of obstacles. All in all, the evaluation could be clearer and more detailed in some aspects. On a side note, it is mentioned that tied f values are broken by *higher* g value. Why are higher g values preferred and not lower ones? Would that not correspond to longer durations?

Needless to say, the paper is slightly overlength. Although the provided examples are nice, it is possible to reduce space here without compromising on their usefulness.

Miscellanous Comments:
- I find it odd to have contiguous line numbers across seperate algorithms
- I also find it odd for the scatter plots in Figure 3 to have different axes lengths. Is there a reason for this?
- There is a broken citation: Halp*re*n (1974) should be Halp*er*n (1974)
- Abstract: "... we introduce algorithms to plan using it." - I found this sentence very hard to parse.

Typos:
- Missing spaces between a word and a following parenthesis (occurs multiple times)
- Page 2: "represent an interval in time"
- Page 3: "three times and a duration" - "time points" would make this more parsable
- Page 4: "equivelent"

---

> ### Author Response · Authors · 2023-05-01
> **Review 2 Response**
>
> Thank you for your helpful comments.  We sincerely apologize for the typos, which we will correct for the camera ready version, including the Halpern citation.  It is our understanding that the page limit is 9 pages and our submission was less than 9 pages by almost half a column, please let us know if our understanding is incorrect.
>
> To address specific points:
> In eqn 5 $\min(\alpha_e,\beta_e)$ is used so that eqn 5 also applies to path ATFs, where $\alpha > \beta$ may be true.  This is discussed in the final paragraph of the first column of page 4, however we agree that this is a point of confusion that should be addressed when eqn 5 is first presented and we will add a description of eqn 5 to the edge ATF subsection which will include a description of why we use the min.
> Your description of RSIPP is accurate, and we agree that it would be useful to be more clear than saying it does not truly solve @SIPP.  We will reword the text to be more precise and make it clear that RSIPP is approximating a solution to @SIPP by repeatedly solving an offline SIPP problem, effectively pretending its runtime is instantaneous.
> We apologize for our oversight in not fully explaining figure 3, your guess and interpretation of the figure are correct, and we will modify the text to explain that.
> We presented figure 3 as three subplots rather than a single plot on a shared axis because putting all three on the same axis made the progression less clear, and we have sufficient space to split out the figure.  If by the `axis being different' you were referring to the scale of the x and y axis being slightly different, we will make the scales the same for the camera ready version.
>
> With regards to the runtime comparison, the 40x is a comparison between the runtime of PEAT solving the @SIPP problem and SIPP solving the offline SIPP problem. So while the algorithms are similar, and the first path to the goal PEAT finds will correspond to the solution found by SIPP the @SIPP problem is much harder than the SIPP problem because we are solving for any start time (up to 4s), instead of a single start time.  We share the reviewers opinion that 40x seems like a lot, as we suggest at the end of that subsection how to improve the runtime of PEAT, or design a better algorithm for @SIPP is an active focus of our current research.  We can add a description of why we believe additional pruning for PEAT is possible.
> In A* it is common to break ties in $f$ in favor of higher $g$, which intuitively represents more complete paths on open, we want to expand the goal first for the final f layer (when $h=0$).  This tie breaking has been found to often reduce the number of expansions needed by the search.

---

> > ### Comment · Reviewer_VyKp · 2023-05-02
> > **Thank you for your Response**
> >
> > Thank you for your response. As a matter of fact, the page limit is indeed 9 pages including references. I apologize for the confusion!
> >
> > > If by the `axis being different' you were referring to the scale of the x and y axis being slightly different, we will make the scales the same for the camera ready version.
> >
> > Yes, it is totally fine to have three subplots, but the different scales in the scatter plots are slightly confusing.
> >
> > > We share the reviewers opinion that 40x seems like a lot, as we suggest at the end of that subsection how to improve the runtime of PEAT, or design a better algorithm for @SIPP is an active focus of our current research. We can add a description of why we believe additional pruning for PEAT is possible.
> >
> > Are there any things you have in mind or have tried out yet? I do not see this as too important since the additional overhead is unavoidable if we truly commit to this setting, but I think this would nevertheless be interesting.

---

### Official Review · Reviewer_SHky · 2023-04-26
**Very clear description and discussion of a previously unstudied problem.**

**Rating:** 9
**Confidence:** 3

**Review:**

### Summary
The paper studies a new problem denoted by any-start-time safe interval
path planning (@SIPP). It complements other research areas in temporal
planning in that it is the first offline planning approach to problems
where the start-time of executing a plan is unknown. The problem is
outlined very clearly and the text is easy to follow. An example is
introduced early and repeatedly used to guide the reader through the
discussion of different theoretical aspects. I really liked reading this
paper and think the authors did a great job in clearly outlining the
problem and their approach to solving it.

An experimental evaluation considers standard pathfinding problems
extended with moving objects to model the changing environment required
for an interesting analysis of @SIPP problems. Certain paths are
sometimes blocked by these moving objects and therefore some actions are
not always applicable. The agent searching for a shortest plan can
instead wait at any position to consider the option of taking that path
once it becomes unblocked again. The overhead of @SIPP algorithms
compared to regular SIPP seems bearable. Furthermore, planning ahead of
knowing the start time is reasonable in scenarios where time is the
critical part of the cost function.

### Minor Comments
- Oftentimes spaces are missing before citations.
- Introduction: "ie." --> "i.e."
- Some parts of "Earliest ATFs for Paths" only became clear once I read
  the example, but I cannot really point out what information I was
  missing.
- "Searching with ATFs", definition of $f$ in parentheses: "($f(n) =
  \alpha(n) + \Delta(n) + h$" --> "($f(n) = \alpha(n) + \Delta(n) +
  h(n)$)"
- I don't understand the following sentence: "We measured the overhead
  of each instance as one hundred times the increase in runtime of ASIPP
  divided by the runtime of SIPP."
- "How long are SIPP plans valid?": something is off with "(confidence
  intervals are all are ..."
- "How long are SIPP plans valid?": "make a @SIPP approach attractive."
  --> "make an @SIPP approach attractive."
- Conclusion: "This work strengths ..." --> "This work strengthens ..."

---

> ### Author Response · Authors · 2023-05-01
> **Review 1 Response**
>
> Thank you for your helpful comments.
> We apologize for the typos, and will correct them for the camera ready version. We will rewrite the definition of overhead to make it more clear.  We will also try and make the concept of path ATFs more immediately understandable.  We will highlight earlier that: the ATF of a path is the composition of the ATF of it's edges, and like a single edge has a certain time the agent will be required to wait until ($\alpha$) and a departure time where the path can no longer be traversed ($\beta$).  There is ample space for us to make these changes while meeting the page limit.

---

### Decision · Program_Chairs · 2023-05-05

**Decision:**

Accept

**Comment:**

We are happy to announce that the paper has been accepted for the workshop, congratulations.

Both reviewers clearly expressed their support for the paper. Please make sure to incorporate the feedback mentioned by the reviewers and fix all typos for the the final version.